# Mapping the Diverse and Inclusive Future of Parkinson’s Disease Genetics and Its Widespread Impact

**DOI:** 10.3390/genes12111681

**Published:** 2021-10-23

**Authors:** Inas Elsayed, Alejandro Martinez-Carrasco, Mario Cornejo-Olivas, Sara Bandres-Ciga

**Affiliations:** 1Faculty of Pharmacy, University of Gezira, Wad Medani P.O. Box 20, Sudan; inaselsayed25@gmail.com; 2International Parkinson Disease Genomics Consortium (IPDGC)-Africa, University of Gezira, Wad Medani P.O. Box 20, Sudan; 3Queen Square Institute of Neurology, University College London (UCL), London WC1E 6BT, UK; alejandro.carrasco.20@ucl.ac.uk; 4Neurogenetics Research Center, Instituto Nacional de Ciencias Neurológicas, Lima 15003, Peru; mario.cornejo.o@incngen.org.pe; 5Center for Global Health, Universidad Peruana Cayetano Heredia, Lima 15103, Peru; 6Molecular Genetics Section, Laboratory of Neurogenetics, National Institute on Aging, NIH, Bethesda, MD 20892, USA

**Keywords:** Parkinson’s disease, genetics, diversity, post-GWAS era, genetic testing, genetics counselling

## Abstract

Over the last decades, genetics has been the engine that has pushed us along on our voyage to understand the etiology of Parkinson’s disease (PD). Although a large number of risk loci and causative mutations for PD have been identified, it is clear that much more needs to be done to solve the missing heritability mystery. Despite remarkable efforts, as a field, we have failed in terms of diversity and inclusivity. The vast majority of genetic studies in PD have focused on individuals of European ancestry, leading to a gap of knowledge on the existing genetic differences across populations and PD as a whole. As we move forward, shedding light on the genetic architecture contributing to PD in non-European populations is essential, and will provide novel insight into the generalized genetic map of the disease. In this review, we discuss how better representation of understudied ancestral groups in PD genetics research requires addressing and resolving all the challenges that hinder the inclusion of these populations. We further provide an overview of PD genetics in the clinics, covering the current challenges and limitations of genetic testing and counseling. Finally, we describe the impact of worldwide collaborative initiatives in the field, shaping the future of the new era of PD genetics as we advance in our understanding of the genetic architecture of PD.

## 1. Introduction

Parkinson’s disease (PD) is a complex neurodegenerative disorder whose prevalence is predicted to increase drastically, being more pronounced in older age people and with variations among sex and ancestry groups [1]. PD is clinically manifested as resting tremor, gait impairment, bradykinesia (slowness of movements), rigidity, and postural instability.

The heritability of PD driven by common genetic variation is estimated to be ~22% and only approximately one third of it has been uncovered with the largest genetic study in the European ancestry population to date [2]. PD is a complex genetic disease, and such heritable genetic variation has a different magnitude of effect, frequency, deleteriousness, and penetrance, so that we can differentiate between rare or common variants, pathogenic deleterious mutations or variants that slightly increase the risk for developing PD, and incomplete versus complete penetrance [3]. The vast majority of patients with PD are diagnosed as sporadic without a clear genetic cause but probably due to the interplay between genetic and environmental risk factors. However, up to 15% of PD patients have a positive PD family history and 5–10% respond to Mendelian inheritance [4]. The last twenty years have witnessed the discovery of recessively and dominantly inherited genes responsible for rare monogenic forms of PD [5]. Well-known, highly penetrant variants causing familial or early onset PD are found within *SNCA*, *VPS35*, *PARKN*, *DJ-1*, and *PINK1* genes. In addition, risk variants with incomplete penetrance have been reported within *GBA* and *LRRK2* [6], as well as 90 risk variants increasing the susceptibility of PD in Europeans and Asian populations [2,7]. Despite the progress made in understanding the genetic basis of PD, current available genetic testing is mostly used on familial and early onset PD cases requiring appropriate genetic counseling.

The discovery of genes and loci associated with PD allows us to redefine the genetic map of the disease, gaining knowledge of potential mechanisms contributing to PD. Understanding PD etiology gives us valuable insights to develop disease-modifying therapies that may stop or slow the progression of the disease. Genome wide association studies (GWAS) have been a powerful tool to better understand how genetics contribute to the risk, progression, and onset of PD [2,8,9]. However, a drawback of all the progress made in understanding PD genetics is that the vast majority of studies have focused on individuals of European ancestry, leading to a gap of knowledge on the likely existing genetic differences among populations.

In this review, we aim to outline the need, benefit, and challenges of exploring the genetic basis of PD across underrepresented populations. We provide an overview of the current state of the field and its applicability in clinical practice, as well as highlight the role of worldwide initiatives in shaping the future of the new era of PD genetics as we advance in our understanding of the genetic architecture of PD.

## 2. Parkinson’s Disease Genetics in Underrepresented Populations: Need for Inclusivity

The recent decade has witnessed unprecedented growth in PD genomics research. This has produced a substantial improvement in therapeutic development. Despite efforts, the effective translation of PD research outcomes into health care optimization has failed at generalizability due to the limited ethnic diversity of studies, since most PD genetic studies have been conducted on populations of European descent and, lately, Asian ancestry populations [2,7]. Such paucity in the representation of ethnically diverse populations, including Africans, South Americans, and indigenous populations, can culminate in a serious disparity in the quality of health care delivered to PD patients [10,11].

The inclusion of ethnic diversity in PD genetics research is essential to improve PD health care in many aspects. First, the under-studying of non-European populations can lead to the underestimation of genetic risk factors specific to that population, which can serve as valuable markers for early disease detection and risk quantification. In addition, including diverse populations can help validate or refute previously identified risk loci in European populations and highlight potential variability in genetic variants’ contributions to PD risk across different populations. An example of these differences can be found in the largest GWAS undertaken in the Asian population, in which associations between the PD phenotype and *GBA* or *MAPT* variants were not found [7].

Moreover, the ethnic diversity in PD research is crucial to improve our understanding of the disease’s biology and pathogenesis. Addressing the variability in the genetic architecture of PD research across populations can help us capture a broader range of genetic and environmental factors implicated in disease development and progression, and tailor the ideal preventive measures and therapeutic interventions accordingly. For instance, these include rare genetic variation with key implications in disease pathogenesis, which could be better highlighted in certain populations while overlooked or completely missed in others because of naturally occurring, population-based variations in allele frequencies [12].

By addressing the variability in genetic architecture and environmental conditions, diversifying PD genetic research can also help us comprehend the interplay between the contributing common variants and environmental factors. This is important since many diagnostic/risk assessment algorithms like polygenic scores are based on the dosage of common variants’ contributions to disease’s risk/pathogenesis identified in European populations, which are not necessarily applicable to non-European populations [5,13]. Generally, translating genetic information derived from studies based on European ancestry populations to other ethnicities with a distinct genetic background and environmental exposure might produce limited gains in the future, or potentially even worse clinical outcomes. This underscores the importance of diversifying PD genetic research, which represents a current global priority.

A better representation of understudied populations in PD genetics research requires addressing, and resolving, all the challenges that hinder the inclusion of these populations (Figure 1). Several factors were found to be responsible for the underrepresentation of non-European populations in PD research. The major recognized challenge limiting the access of populations living in low and middle-income countries (LMICs) to PD research is the lack of funding and infrastructure. Hence, allocating funds to support genetics studies in PD in these countries can help to improve the accessibility of these populations to research [14]. Besides the financial and logistic limitations, the availability of a trained scientific workforce is another major challenge in LMICs. Fortunately, establishing training programs targeting the scientific workforce in under-developed countries is currently more feasible with the aid of virtual tools and technologies [15]. To address this to some degree, the Global Parkinson’s Genetics Program (GP2, https://www.gp2.org/, accessed on 20 September 2021) [16] has recently established a virtual center of excellence with resources and expertise to serve the training needs across these populations.

In addition, the lack of motives to participate in genetics research either due to low awareness about research benefits or negative perceptions about research or towards research procedures, especially invasive procedures like blood sampling, can also significantly limit the inclusion of certain populations in PD genetics research [17,18,19]. This combined with potential restricting cultural and/or religious barriers were found to limit the participation of understudied populations both in LMICs as well as minorities living in high-income countries [19]. Organizing educational programs in targeted communities to improve populations’ awareness about PD genetics research and reduce cultural stigma, combined with developing policies and regulations to protect participants’ confidentiality and safety is essential to guarantee the better engagement of targeted populations in genetic studies [20].

When thinking about genetic research in non-European populations, there are some limitations to take into account. One of them is the variation in linkage disequilibrium patterns and haplotype structure between ethnically diverse populations that can complicate GWAS imputation while using genotyping panels designed for European populations in other populations [13,21]. Furthermore, the heterogeneity of certain populations, particularly African and Latino populations, where complex ancestry admixture exists, represents another major challenge [13].

Fortunately, in the last few years, several national and international endeavors have been launched to improve population diversity in PD research. One of the prominent international initiatives supporting ethnic diversity in PD genetic research is The Global Parkinson’s Genetics Program (GP2) from the Aligning Science Across Parkinson’s (ASAP) initiative [16]. Aiming to enhance PD genetics research and population diversity to generate comprehensive, reproducible, and accessible data, GP2 has devoted significant resources to establish research infrastructure and train researchers in PD research around the globe. An example of its commitment is the GP2 Black and African American Connections to Parkinson’s disease study (BLAAC PD) launched in 2021. This project targets one of the most underrepresented in neurodegenerative disease studies, which are the African American and Black American populations [19]. Similarly, The International Parkinson Disease Genomics Consortium-Africa (IPDGC-Africa) and Latin American Research Consortium on the Genetics of PD (LARGE-PD) have established PD research programs targeting underrepresented populations in Africa and Latin America respectively [22,23,24]. Besides enhancing the representation of understudied populations, these initiatives aim to improve PD research facilities, train the local workforce, and engage the communities through promoting research-supporting concepts and alleviating negative notions [16,22]. Such endeavors are expected to improve diversity in PD genetics research and warrant equity in medical services provided for PD patients around the globe.

## 3. Parkinson’s Disease Genetics in the Clinic: Interpretation of Genetic Testing and Genetic Counseling. Challenges and Limitations

Genetic testing is mostly defined as DNA-based testing performed within a medical context for health care purposes with the intention to counsel individuals or families on the risk of diseases or implications to health and life decision-making. Depending on the specific purpose of the genetic test it can be diagnostic or predictive [25]. Genetic testing is often expensive, time-consuming, and not necessarily accessible in some countries. Diagnostic genetic testing, when positive, not only stops expensive diagnostic tests, but also has therapy implications and allows appropriate counseling on severe life decisions. On the other hand, a negative report might reorient a differential diagnosis or lead to future reassessment and further investigation [25].

Genetic testing and appropriate counselling for complex disorders like PD has been rapidly evolving, however there are many aspects to be considered when requiring a genetic test in PD. The increasing knowledge in the PD genetics field and the advent of technology have revolutionized genetic testing and counseling over the past decades. The technology used in genetic testing has rapidly evolved from single-gene approaches to next-generation sequencing, including exome and genome sequencing. As a result, genomic data for diagnostic purposes has been generated at a large scale and in an unprecedented manner, often requiring high capacities and resources for a clinician to interpret it [26]. The increasing application of genetic testing in clinical practice has been related to the decline of genotyping and sequencing costs. However, downstream requirements for genomic interpretation still limits its broader use in complex disorders like PD [27].

Since the identification of missense variants in the *SNCA* gene in 1997 [28], genetic mutations in about twenty genes have been described (Table 1), with at least six of them showing consistent evidence for causality [29]. Many genetic variants in *SNCA*, *VPS35*, *PRKN*, *LRRK2*, *PINK1*, and *DJ1* among other genes, have been consistently linked to monogenic PD forms representing approximately 5% of all PD cases [30,31]. However, incomplete penetrance, often seen in *LRRK2* and *GBA* variants, implies limited use for establishing individual risk in clinical practice. Additionally, a relevant situation to mention relates to these two genes harboring one or few founder mutations that are particularly frequent in certain populations.

*LRRK2* and *SNCA* mutations are often screened in the presence of family history and suspicion of monogenic autosomal dominantly inherited PD. Affected PD patients carrying *LRRK2* mutations have been reported worldwide with higher frequencies among Ashkenazi Jewish and Tunisian Barber populations [32], and lower frequencies among East Asians and Latinos with high Amerindian ancestry [33,34]. *LRRK2*-*G2019S* and ROC (Ras of complex) domain variants (R1441G/C/H) are among the most common variants associated with PD. Despite the fact that motor symptoms and responses to levodopa do not differ from idiopathic PD, some studies suggest a lower frequency of non-motor symptoms and mild cognitive impairment [35]. Age-dependent penetrance has been consistently demonstrated, with higher rates within *LRRK2*-*G2019S* carriers [36]. Genetic testing among putative *LRRK2* carriers can be useful from the patient and research perspective, as clinical trials of *LRRK2* kinase inhibitors have started showing promising results as the first personalized therapies for monogenic *LRRK2* patients [30,37]. Missense and copy number variants (duplications and triplications) in the *SNCA* gene have been linked to monogenic autosomal inherited PD [38]. Clinical phenotypes of *SNCA* carriers are quite variable but often severe, with some *SNCA* mutations and rearrangements being related to a higher frequency of cognitive impairment, psychosis, and depression [39,40]. Thus, genetic diagnostic panels that include *LRRK2* and *SNCA* should be considered when affected individuals with autosomal dominant familial PD are seen in clinics. 

Biallelic rare variants within the *PRKN*, *PINK1* and *DJ-1* genes are consistently associated with early onset recessive PD. These three genes encode proteins sharing a common pathway, mitochondrial quality control and regulation [41]. Main clinical features related to variants within these genes are mostly consistent with early onset disease with slower progression, excellent Dopa response, frequent dystonia, dyskinesia, and uncommon cognitive decline [42]. Genetic testing for *PRKN*, *PINK1*, and *DJ-1* in familial forms of PD with recessive patterns and in early onset PD cases might be considered on diagnostic and therapy algorithms. Early onset PRKN PD cases usually have a prolonged and consistent response to low doses of levodopa, however they tend to develop levodopa-induced dyskinesias as well as compulsive disorders with the use of dopamine agonists [43]. Other treatment options including DBS have demonstrated positive results in selected cases [44].

Genetic testing and counseling for PD common risk variants is hard to interpret for individual cases in clinical practice. GWAS studies have nominated potential susceptibility factors in *LRRK2* and *SNCA* linked to sporadic PD [3]. *GBA*, the coding gene for glucocerebrosidase, is the most common genetic factor for developing PD and an important risk factor for other synucleinopathies for which multiple clinical trials are ongoing. Heterozygous variants in *GBA* are present in up to 15–20% of PD patients in certain populations and bear a higher risk of non-motor features, such as cognitive decline and dementia [45,46]. The consistent association of *GBA* with PD risk contrasts with the large number of *GBA* carriers who do not develop PD given the low penetrance of this gene [47,48]. 

On the other hand, there is controversial evidence suggesting risk conferred by heterozygous *PRKN* and *PINK1* variants in PD etiology. Large-scale studies systematically interrogating *PINK1* variants failed to confirm its role as risk factor for PD [49]. Given the lack of replication and controversial findings across research studies, genetic testing seeking this specific heterozygous variation is not recommended.

Regular genetic testing for sporadic late-onset PD is not currently recommended as standard clinical practice. PD is considered a complex disorder with the coexistence of a genetic predisposition together with variable environmental exposure [50]. Sporadic PD is the most common form of neurodegenerative Parkinsonism, representing the vast majority of cases seen in regular clinical practice. Despite the tremendous advances in understanding the genetic architecture of PD, it is still challenging to explore individual genetic risk in sporadic forms. Novel multi-OMIC approaches are being investigated to predict PD risk, including polygenic risk stratification and multimodal data integration.

Genetic testing for PD should be performed within appropriate genetic counseling approaches depending on the individual clinical profile. Genetic testing might be highly valuable in the presence of a positive family history, early onset PD, or specific high-risk ancestry like Ashkenazi Jewish [51]. While there is strong evidence for a potential use of genetic testing for monogenic PD, not only for diagnostic purposes, but also for precision medicine decisions, there are still significant limitations including the existence of variants with variable penetrance, variants of uncertain significance, and the presence of other susceptibility genetic factors [52]. Given the complexity of PD, it is strongly recommended to discuss the benefits and limitations of genetic testing during pre-test counseling sessions, including the risk of privacy loss and discrimination [53]. Since direct-to -consumer testing for common variants of *LRRK2* and *GBA* genes is currently available, care providers must be trained in genetic counseling to address consultants’ concerns. Comprehensive genetic education and training of clinicians and patients together with efforts to promote universal access to genetic services are needed to massively translate PD genetic testing into clinical practice across the globe [54].

## 4. The New Era of Parkinson’s Disease Genetics: Increasing Knowledge about Disease Etiology

The new era of PD genetics holds promise. Genetic studies conducted in underrepresented populations have started to emerge [55,56,57] as well as consortium setups [58] (GP2, https://www.gp2.org/, accessed on 20 September 2021) with programs strongly focused on increasing diversity in PD research so that the applications of genetic discoveries can be extrapolated to the entire population. As a result, an important analytical approach in our field will be the implementation of trans-ethnic GWAS meta-analysis, such as GWAMA [57] and MANTRA [59], that will allow us to combine genetic information from different ancestries to further delineate the etiology of this complex disease. GWAS data from non-European samples is considerably increasing, ensuring higher statistical power to improve fine-mapping strategies by leveraging the LD structure from different populations [59]. A considerable improvement in the fine-mapping resolution when studying data across highly ancestral heterogeneous samples has been shown as opposed to fine-mapping based on European ancestry only data [60].

When it comes to further exploring nominated loci from GWAS, the altered molecular pathways contributing to the phenotype of interest may be diverse [61]. Additionally, among the nominated loci, it is usually challenging to detect the causal variant underneath the peak, often masked by other non-causal alleles falling within the same haplotype block as a result of the underlying LD structure. In this context, the tuning of genotyping approaches as well as the development and implementation of novel bioinformatics tools are of paramount importance. On the one hand, some novel genotyping platforms, such as the recently created Neuro Booster array (unpublished manuscript), are focused on a wide SNP coverage, including more than 1.8 million variants (as compared to the roughly 400,000 variants from previous arrays [62,63]), and a custom content of approximately 95K neurological disease related variants. On the other hand, a wide range of publicly available and useful data science approaches allows us to interpret GWAS outcomes and further dissect potential loci. Fine-mapping methods represent a means to come up with the likely causal variant from a specific locus for a given phenotype and to determine the functional implications of such loci [64]. In very few situations, the causal variant will be within the locus of interest, affecting the protein conformation if the mutation is coding. Most likely, a causal genetic variant can be found within a non-modifiable or regulatory region, resulting in the dysregulation of the gene product of interest. Colocalization methods allow us to explore whether GWAS studies share a common genetic causal variant with tissue level and cell-state-specific expression quantitative trait loci (eQTL datasets), allowing us to link GWAS single nucleotide polymorphisms (SNPs) to the regulation of gene expression [64]. Moreover, functional fine-mapping methods give us insight into the putative epigenetic signatures of GWAS nominated loci, such as DNA methylation or histone modification of regulatory elements, as well as the formation of chromatin loops [65].

The identification of culprit variants affecting PD risk may be possible by the implementation of state of the art high-throughput long read sequencing technologies. Causal variants do not necessarily have to be SNPs, but can also be more complex genomic variation, such as repeat expansions or structural variants which are easily overlooked in short-read sequencing, and/or technologically challenging to genotype due to repetitive sequences or high GC content. PD studies looking at non-SNV variation are starting to emerge [66].

Undertaking integration of different level data (i.e., clinical data, genetics, transcriptomics, proteomics, and metabolomics) can be challenging and costly. Fortunately, it is worth highlighting that some frameworks that facilitate the process for post-GWAS analyses are available [67]. These platforms include large integrated biological datasets, making the automatization of concrete and parallel analyses possible, easing reproducibility and transparency. As we move forward, standardization and harmonization of datasets, as well as automating data processing is key. An example of this is GenoML (https://genoml.com/, accessed on 20 September 2021) which enables automatic machine learning in genetic studies and has been widely applied in the PD genetics field [68,69].

Overall, post-GWAS analyses are focused on approaches to prioritize molecular pathways and promising targets for biomarkers and drug development. By discovering and validating potential findings in independent cohorts, we can nominate pathways to be assessed in cell lines and animal models or build up networks. Moreover, novel datasets for PD genetics research are currently being made public resources to the research community. The Foundational Data Initiative for Parkinson’s Disease (FOUNDIN-PD) [70] is an international, collaborative, and multi-year project, aiming to produce a multi-layered molecular dataset in a large cohort of 95 induced pluripotent stem cell (iPSC) lines at multiple time points during differentiation to dopaminergic (DA) neurons (https://www.foundinpd.org/#Foundinpd, accessed on 20 September 2021).

## 5. Future Perspectives

Over the last 20 years, in many ways, genetics has been the engine that has pushed us along on our voyage to gain knowledge about PD etiology. As we move forward, shedding light on the genetic architecture contributing to PD in non-European populations is essential and will provide novel insights regarding the generalised genetic map of the disease. This is a major commitment and a significant step forward for our field in an effort to understand how the basis of disease varies across populations.

We envisage that we will continue increasing the number of known genetic risk loci disease-causing mutations for the complex and variable manifestations of PD. Our field will keep investigating risk loci to saturation, genetic modifiers of disease, and genetically defined disease subtypes. In terms of genetic players underlying PD etiology, we anticipate that our field will expand our understanding of structural and repeat variability involved in disease through the application of long-read sequencing, which so far has been relatively difficult to explore using traditional genome sequencing methods. 

However, in the future, our field will not just strive to improve our understanding of the role genetics plays in PD on a global scale, but to also make that understanding actionable. Worldwide initiatives will be key to the creation of publicly available resources for the scientific research community [16]. Multimodal data integration will facilitate translation of genetic maps to mechanisms and will improve our ability to develop more accurate models of disease prediction and prognosis.

It is not enough to just make data available to the wider PD research community, we must train the next generation of scientists. Training individuals with exceptional drive and talent will be key to success. The future of PD genetics ultimately aims to inform biology, improve disease modeling, promote target prioritization, inform trial design and efficiency, and develop therapeutic strategies matching patients to specific treatments.

## Figures and Tables

**Figure 1 genes-12-01681-f001:**
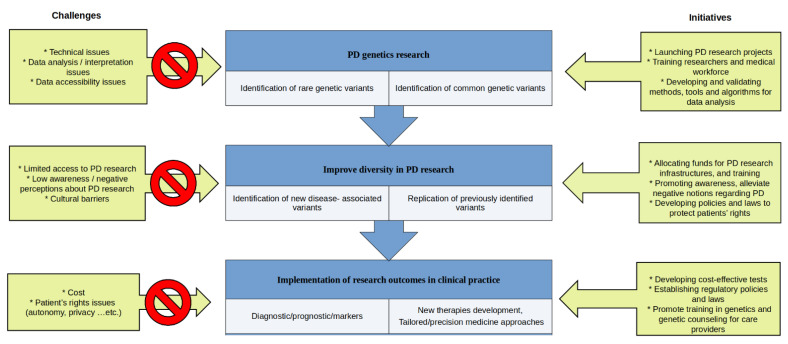
The Parkinson’s disease genetics path: From research to clinics. Parkinson’s disease (PD).

**Table 1 genes-12-01681-t001:** List of genes reported to be linked with Parkinson disease.

Gene	Year of Discovery	Reported Variants	Frequency	Inheritance	Confidence as a PD Gene
*SNCA **	1997, 2003	Missense or multiplication	Very rare	Dominant	Very high
*PRKN **	1998	Missense or loss of function	Rare	Recessive	Very high
*UCHL1*	1998	Missense	Unclear	Dominant	Low
*PARK7 **	2003	Missense	Very rare	Recessive	Very high
*LRRK2 **	2004	Missense	Common	Dominant	Very high
*PINK1 **	2004	Missense or loss of function	Rare	Recessive	Very high
*POLG*	2004	Missense or loss of function	Rare	Dominant	High
*HTRA2*	2005	Missense	Unclear	Dominant	Low
*ATP13A2 **	2006	Missense or loss of function	Very rare	Recessive	Very high
*FBXO7 **	2008	Missense	Very rare	Recessive	Very high
*GIGYF2*	2008	Missense	Unclear	Dominant	Low
*GBA **	2009	Missense or loss of function	Common	Dominant (incomplete penetrance)	Very high
*PLA2G6 **	2009	Missense or loss of function	Rare	Recessive	Very high
*EIF4G1*	2011	Missense	Unclear	Dominant	Low
*VPS35 **	2011	Missense	Very rare	Dominant	Very high
*DNAJC6*	2012	Missense or loss of function	Very rare	Recessive	High
*SYNJ1*	2013	Missense or loss of function	Very rare	Recessive	High
*DNAJC13*	2014	Missense	Unclear	Dominant	Low
*TMEM230*	2016	Missense	Unclear	Dominant	Low
*VPS13C*	2016	Missense or loss of function	Rare	Recessive	High
*LRP10*	2018	Missense or loss of function	Unclear	Dominant	Low
*NUS1*	2018	Missense	Unclear	Recessive	Low

*** Gene of PD clinical significance adapted from Blauwendraat et al., 2019. Parkinson’s Disease (PD).

## Data Availability

Not applicable.

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
