# Peer review of "Mapping the Diverse and Inclusive Future of Parkinson’s Disease Genetics and Its Widespread Impact"

_genes, 2021, doi:10.3390/genes12111681_

Round 1
Reviewer 1 Report
Comments:
This is a comprehensive review which aims to outline the need, benefit, challenges and limitations of exploring the genetic basis of PD across underrepresented populations. Overall the review is well written and recent bibliography/publications are included. However, there are some points to be improved.
Major changes
- A comprehensive table to list all the different mutations/variations will be very helpful for the reader and will improve what the review is trying to highlight (like you are discussing in lines 190-229), including type of study, gene, variation/mutation, ethnicity, symptoms, homozygous/heterozygous, frequency etc.
Minor changes
- Add reference for Global Parkinson’s Genetics Program, GP2 (line 117).
- Use table to show the different projects GP2, BLAAC PD etc, including details such as sample size and ethnicity of the cohorts (line 138-156).
- Line 168: add reference.
- Line 183, PARKN > PRKN
- Line 186, GBA > italics
- Line 194 ROC > ROC (Ras of complex)
- Part 5. Future perspectives; references are missing from this part
- Line 333; there is an extra space before the “In terms…”
- It seems to be a problem with the numbering of your refences that starts at number 6 line 365 (the refence number it appears for second time)
Author Response
Comments:
This is a comprehensive review which aims to outline the need, benefit, challenges and limitations of exploring the genetic basis of PD across underrepresented populations. Overall the review is well written and recent bibliography/publications are included. However, there are some points to be improved.
Response: We thank Reviewer 1 for the positive feedback
Major changes
- A comprehensive table to list all the different mutations/variations will be very helpful for the reader and will improve what the review is trying to highlight (like you are discussing in lines 190-229), including type of study, gene, variation/mutation, ethnicity, symptoms, homozygous/heterozygous, frequency etc.
Response: We thank Reviewer 1 for this suggestion and have included a table listing all the different PD genes accordingly (Table 1)
Minor changes
- Add reference for Global Parkinson’s Genetics Program, GP2 (line 117).
Response: We have incorporated the reference (PMID: 33513272)
2. Use table to show the different projects GP2, BLAAC PD etc, including details such as sample size and ethnicity of the cohorts (line 138-156).
Response: Since these are ongoing initiatives that include multiple projects, the estimated sample size dynamically changes according to the progress. We thought it would be more useful to provide project references so the interested reader can find a link to the project’s specific websites where the updated information will always be there.
3. Line 168: add reference.
Response: Reference added according to reviewer’s comment
4. Line 183, PARKN > PRKN
Response: Line edited according to the reviewer’s advice.
5. Line 186, GBA > italics
Response: Line edited according to the reviewer’s advice.
6. Line 194 ROC > ROC (Ras of complex)
Response: Line edited according to the reviewer’s advice.
7. Part 5. Future perspectives; references are missing from this part
Response: We apologize for this oversight and have included references accordingly.
8. Line 333; there is an extra space before the “In terms…”
Response: Line edited according to the reviewer’s advice.
9. It seems to be a problem with the numbering of your refences that starts at number 6 line 365 (the refence number it appears for second time)
Response: Line edited according to the reviewer’s advice.
Reviewer 2 Report
The review paper “Mapping the diverse and inclusive future of Parkinson’s disease genetics and its widespread impact” by Inas et al., mainly review the current and future plans of “The Global Parkinson’s Genetics Program (GP2)”.
Although many PD-causative mutations and risk loci have been described, the majority were identified by studying individuals of European ancestry.
In order to enrich our understanding of the complex genetics of PD, there is a real need to study many other different PD populations worldwide. This goal will hopefully be achieved by the GP2 project.
This review gives clear background and detailed picture of the goals and structure of PG2, and publishing this information to the readers of the special PD-GENES issues is important. The review is clearly written and I have no major comments
Author Response
Response: We thank Reviewer 2 for the positive feedback